# Metastases to Meningiomas: A Comprehensive Literature Review Including Mediating Proteins

**DOI:** 10.3390/cancers14235877

**Published:** 2022-11-29

**Authors:** Mahlon D. Johnson

**Affiliations:** Department of Pathology, Division of Neuropathology, University of Rochester Medical Center, 601 Elmwood Ave. Box 626, Rochester, NY 14623, USA; mahlon_johnson@urmc.rochester.edu; Tel.: +1-(585)-276-3087; Fax: +585-273-1027

**Keywords:** meningioma, tumor-to tumor metastasis, mesothelin, cell adhesion molecules

## Abstract

**Simple Summary:**

Metastasis from recognized, or occult, malignancies to a co-existing meningioma can present diagnostic challenges. This is, in part, do to the potential histological similarity of the metastasis to the recipient meningioma tissue. Moreover, the discovery of the metastases, particularly from an occult tumor, has important therapeutic implications for managing these scenarios. The review was undertaken to determine the types of neoplasms that spread to meningiomas and identify what histologic subtypes of meningiomas that are at greatest risk of receiving metastases. The histologic challenges of identifying small metastases in meningiomas are discussed. The molecules and mechanisms by which they facilitate this tumor-to-tumor spread are discussed.

**Abstract:**

Approximately 5–15% of solid tumors metastasizing to the central nervous system metastasize to the leptomeninges. Less common, is metastasis to leptomeningeal meningiomas. These are primarily carcinomas of the breast and lung. Awareness of this phenomenon is critical to the evaluation of meningiomas, especially since the metastases may be the first indication of an occult tumor elsewhere in the body. Lack of clear demarcation between the metastasis and meningioma parenchyma, as well as histological features similar to the meningioma, may hinder recognition. The mechanisms underlying metastases anchoring and spread along the leptomeninges are not established. However, several cell adhesion molecules are thought to contribute to this phenomenon. E cadherin is a cell adhesion molecule present in meningioma cells. Binding to endothelium by adhesion molecules such as ICAM, B1 integrin, P-selectin, PECAM-1, CXCL12 and SDF-1 have also been proposed as part of the mechanisms underlying breast carcinoma metastases. In addition, the leptomeninges and meningiomas express mesothelin that acts as an anchoring protein coupling with mucin-16. Consequently, metastatic tumor cell mucin and mesothelin may also facilitate the anchoring of metastases to meningiomas.

## 1. Introduction

Approximately 5–15% of solid tumors spreading to the central nervous system metastasize to the leptomeninges. Less common, is metastasis to leptomeningeal meningiomas [1,2,3]. The biology and clinical features of these metastases is likely different from spread to the brain parenchyma per se since the leptomeninges are outside the blood–brain barrier, bathed in cerebrospinal fluid, and heavily vascularized. To understand this process, literature searches for articles in English in PubMed were reviewed, using terms “meningioma”, “metastasis”, “carcinoma”, “tumor-to tumor metastases”, meningioma and cell adhesion molecules and meningioma and mesothelin as key words. To date, approximately 75 cases of metastasis to meningiomas have been reported as small collections or case reports. Cases of meningiomas with tumor-to tumor metastases seen in consultation with the author, at the University of Rochester School of Medicine, were also reviewed and illustrated.

## 2. Epidemiology

The metastases were primarily carcinomas of the breast [2,3,4], adenocarcinoma of the lung [5,6,7,8,9,10], squamous cell carcinoma (Table 1) [11].

## 3. Histopathology

Awareness and screening for this phenomenon is critical to overall patient care since the discovery of a metastases may be the first sign of an occult tumor elsewhere in the body. However, identification of an intra-meningioma metastasis is potentially challenging since they may lack clear demarcation from the surrounding meningioma parenchyma, be only a small focus, and have histological features similar to the meningioma’s (Figure 1 and Figure 2). The lack of distinct borders between the metastasis and meningioma may also obscure their presence. For example, isolated small cell carcinoma metastases may resemble “small cell change” in a WHO grade 2 or 3 meningioma especially if the resected meningioma is not submitted in its entirety.

As an example, we received tissue from a 62-year-old female with headaches and left body tingling. She had anxiety, obesity, and tobacco use. Her cranial nerve exam was normal. Her neurologic exam is unremarkable with intact sensation despite subjective complaints of dysesthesia. Computed tomography (CT) of the head showed an inconspicuous right hemispheric mass causing mass effect and midline shift. The most likely diagnosis was thought to be a meningioma given the possibility of being dural-based, Additional considerations include low-grade glioma and primary CNS lymphoma. However, these were considered less since likely since be a primary malignancy given her cancer screenings had been unremarkable. Nonetheless, subsequent CT scans identified pulmonary nodules. The dural tumor was excised and found to be a WHO grade 2 meningioma. Within it, a small focus of small cell carcinoma with an immunophenotype favoring small cell carcinoma. Biopsy of the lung lesion revealed small cell carcinoma (Figure 1).

Foci of a plasma cell neoplasm may suggest a lymphoplasmacytic meningioma rather than a meningioma with a metastasis from an occult plasma cell neoplasm (Figure 2).

For example, we were asked to evaluate a dural mass in a 93-year-old male with a history of multiple myeloma, HTN, and neuropathy His current illness had been complicated by compression fractures and a cast nephropathy. He developed a transient right facial droop and word finding difficulties.

CT of the head showed interval increase in size of meningioma with significant edema involving the left frontal/parietal lobes, no midline shift or herniation. MR of the head showed a left frontal meningioma. He was felt to have a seizure related to the meningioma, although EEG did not show epileptiform discharges.

Resection revealed a meningioma with tumor-to-tumor plasma cell neoplasm.

Similarly, in consultation, a clear cell carcinoma metastasis to a clear cell meningiomas may potential be overlooked when the borders are indistinct, the meningioma has variably brisk mitoses and focus is small. Usually mitotic activity and histologic features belie the presence of a second tumor. Immunohistochemistry resolves the issue and alerts the referring pathologist to the metastaic tumor. The recognition of the metastasis is important since it may be the first indication of an occult extra-axial malignancy or documentation for staging, of a distant metastasis (Figure 3).

For example, in one case 66-year-old female was evaluated for new onset expressive aphasia. On exam she was awake and alert but not oriented to time but could follow most simple commands and answers questions without difficulty. She read the first two and last sentence on the stroke card fluently. However, she had difficulty naming objects. Her cranial nerves were largely intact. Her bulk, tone, and strength were normal throughout. There was no pronator drift and no abnormal movements. Sensation and vibration were intact. Her finger to nose and heel to shin coordination were intact. Reflexes were 2+ throughout the upper and 3/3 patellar and 2/2 Achilles reflexes.

An EEG identified findings consistent with increased neuronal dysfunction in the left temporoparietal region. This was in the context of a more, diffuse, nonspecific mild encephalopathy. CT scans of the head revealed a pterional, skull base mass with surrounding edema with slight mass effect and midline shift-possibly with necrosis. MRI of the head identified a left middle cranial fossa mass. MRI identified a slight mass effect and midline shift. There also appeared to be more necrosis within this broad-based extra-axial mass lesion.

Her past medical history included a recent evaluation for pain in her ribs and back. That work up included computed tomography scans of the chest that demonstrated a mass in right lower lobe of the lung as well as hypermetabolic right hilar and mediastinal lymph nodes and several bony lytic lesions. MRI of the spine found a compression fracture at T7 and lesions at T3 and L4 suspicious for metastasis. A subsequent CT guided lung biopsy identified adenocarcinoma with *ALK* gene rearrangement. She underwent an excision of the dural mass. Analysis revealed a WHO grade 2 meningioma with intratumoral metastatic adenocarcinoma (Figure 3) [12].

## 4. Molecular Findings

There is no apparent predilection of meningioma subtypes to increased risk of tumor-to-tumor metastasis [1,13]. Nonetheless, features of the metastasis may facilitate the process. In particular, hormone receptor expression may increase the risk of metastasis to a meningioma [14,15]. The role of steroid receptors in meningiomas with metastatic breast carcinoma is noteworthy. Meningiomas are twice as common in women as in men suggesting a hormonal component to the pathogenesis of the tumors with progesterone receptors [14,15]. Approximately 83% of meningiomas have progesterone receptors and 8 to 30% have estrogen receptors [15]. Eight-six percent and 73% of breast carcinomas have progesterone and estrogen receptors, respectively [14]. Women with breast cancer also reportedly have twice the incidence of meningiomas as women in general [14,15]. At autopsy, women with advanced breast carcinoma have meningiomas in approximately 1.2% of cases. Nonetheless, in patients over 67 years of age, incidental meningiomas discovered at autopsy have been found to be slightly more common in men and have an incidence of 2–3% [16].

Vascularity is high in a meningioma may have similar density meningothelial and fibrous. The parenchymal vascularity may be similar between low grade tumors with single or multiple feeders. Atypical and malignant meningiomas have more vascularity. This extensive vascularity in meningiomas per se may be another factor facilitating metastasis to meningiomas. The angiogenesis and peritumoral edema may be facilitated, in part, by increased expression of vascular endothelial growth factor (VEGF) promoting vascular proliferation [1,17].

The mechanisms underlying metastases to meningiomas are not established. In other tissues, adhesion and penetrance of metastases to and between endothelial cells requires binding to adhesion molecules on endothelium and sequential activation of proteases there. A number of cell adhesion molecules are likely to be important. E cadherin is a cell adhesion molecule present in meniningioma cells and many metastases and appears to facilitate adhesion to endothelium [18,19]. Meningiomas with metastases are more likely to express this molecule than meningiomas overall. Cell adhesion molecules such as ICAM, B1 integrin, P-selectin, PECAM-1, CXCL12 and SDF-1 have also been proposed as part of the mechanisms underlying breast carcinoma metastases [20,21,22]. ICAM expression in meningiomas may also facilitate adhesion of metastases to meningioma blood vessels [2].

Additional genes expressed in carcinoma cells likely facilitate carcinoma cell penetrance of the vascular basement membrane in blood vessels walls. The urokinase form of plasminogen activator (uPA) is released by many tumor types at high levels. This displaces normally low level tissue uPA inhibitors resulting in binding of uPA to its receptor and activation of plasmin [23,24]. Plasmin activation increases activation of numerous proteases such as metalloproteases [25]. In highly invasive CD44 expressing breast carcinomas, tumor cells also express a hyaluronic acid binding protein that activates matrix metalloproteases (MMP) 2 and 9 on cell membranes [23,24]. MMP-2 and MMP-9 are especially effective in degrading collagen type IV, a prominent component of vascular basement membranes. Thus, metastases activate a number of proteins that facilitate penetration of the extensive vascular pericellular collagen and hyaluronic acid in the vascular basement membranes of meningiomas.

Transmembrane mucins such as MUC1 and MUC16 are also thought to facilitate the metastasis of many carcinomas including pulmonary adenocarcinomas [26,27]. These bind to a number of membrane proteins such as mesothelin in recipient tissues [26,27]. The human mesothelin gene encodes a 71 kDa precursor protein including a 31-kDa NH2-terminal fragment, megakaryocyte-potentiating factor, which is cleaved and released from the cell to produce the 40 kDa carboxyl, membrane bound mesothelin [28,29]. Mesothelin has been found in the pleura, pericardium, peritoneum and on surface epithelium of some tumors. It is also overexpressed in mesotheliomas, pancreatic adenocarcinomas and squamous cell carcinomas of the head, neck, lung and esophagus, [29]. Recently it has been shown that mesothelin binds MUC16 (originally designated CA125), another extracellular transmembrane protein thru a high affinity, N-glycan dependent interaction [26,30]. In the peritoneum, mesothelin serves as an anchoring site for metastatic adenocarcinoma cells by binding MUC16 [27,29]. This facilitates binding of ovarian carcinomas promoting ovarian carcinomatous peritonitis [27,29]. Other mucins may interact similarly. MUC1, which may also bind to mesothelin, is widely expressed in malignancies and is associated with metastases and invasion [26,31]. In animal models, antibodies to mesothelin block ovarian carcinoma seeding to the peritoneum [26]. Because a number of adenocarcinomas express MUC16, and adenocarcinomas represent one of the most common carcinomas metastasizing to the leptomeninges, it is conceivable that an interaction with meningioma mesothelin and mucins facilitates anchoring by adenocarcinoma metastasis to the leptomeninges and meningiomas In the peritoneum, mesothelin serves as an anchoring site for metastatic adenocarcinoma cells by binding MUC16 [27,29]. This facilitates binding of ovarian carcinomas promoting ovarian carcinomatous peritonitis [27,29]. Other mucins may interact similarly. MUC1, which may also bind to mesothelin, is widely expressed in malignancies and is associated with metastases and invasion [26,31]. In animal models, antibodies to mesothelin block ovarian carcinoma seeding to the peritoneum [26]. Because a number of adenocarcinomas express MUC16, and adenocarcinomas represent one of the most common carcinomas metastasizing to the leptomeninges, it is conceivable that an interaction with meningioma mesothelin and mucins facilitates anchoring by adenocarcinoma metastasis to meningiomas [28,32] (Figure 3).

The function of mesothelin is not established but it may act as a binding site for transmembrane mesothelin and mucins expressed by tumor cells [26,27,29]. Mesothelin, is a proposed adhesion molecule that binds itself and MUC16 in the human leptomeninges [28,32].

Adhesion and penetrance of metastases to and between endothelial cells requires binding to adhesion molecules on the endothelium. A number of cell adhesion molecules are likely to be important. Additional genes expressed in carcinoma cells likely facilitate cell penetrance of the vascular basement membrane thru activation of metalloproteases that degrade collagen and hyaluronic acid. Carcinoma transmembrane mucins such as MUC16 have high affinity for mesothelin expressed in the leptomeninges and meningiomas.

Treatment of meningiomas that contain metastases depends on the circumstances and is addressed by others in this issue. Nonetheless, the discovery of an occult metastasis without known systemic disease would, of course, prompt an extensive search for the primary tumor. Treatment of the systemic cancer would depend on the type of malignancy and staging. Resection of meningiomas with a suspected metastasis might be planned to limit leptomeningeal seeding [1]. Meningiomas with a metastasis extending from the meningioma to the skull or brain would require more extensive resection margins along with radiotherapy. Subsequent treatment would depend on the other extent of resection, grade and the clinical variables of tumor type and location of other extra-axial tumors [33,34]. Meningiomas grow outside the blood–brain barrier. Consequently, penetrance of any chemotherapy to a partially resected or inoperable meningioma with a metastasis would likely be different than in cases of brain parenchymal metastasis.

## 5. Conclusions

Metastasis of extra-axial tumors to meningiomas is a rare event but may be the first indication of an occult peripheral tumor. Metastases may reflect expression of select groups of adhesion molecules on metastatic tumor cells and binding proteins such as mesothelin in the leptomeninges and meningiomas. Recognition of this occurrence is critical to the management of both tumors. Blocking such interactions may represent an additional adjunctive therapy. A number of cell adhesion molecules are likely to be important. E cadherin is a cell adhesion molecule present in many metastases and appears to facilitate adhesion to endothelium [18,19]. Cell adhesion molecules such as ICAM, B1 integrin, P-selectin, PECAM-1, CXCL12 and SDF-1 may also contribute to carcinoma metastases [20,21,22]. Metalloproteases facilitate carcinoma cell penetrance of the vascular basement membrane in blood vessels walls. The urokinase form of plasminogen activator (uPA) is released by many tumor types and facilitates activation of plasmin [24]. Plasmin activation, in turn, increases activation of metalloprotease [25]. In highly invasive CD44 expressing breast carcinomas, tumor cells also activate MMP-2 and MMP-9 resulting in degradation of collagen type IV in vascular basement membranes. Thus, metastases activate a number of proteins that facilitate penetration of the extensive vascular pericellular collagen and hyaluronic acid in the vascular basement membranes of meningiomas.

Transmembrane mucins such as MUC1 and MUC16 are also thought to facilitate the metastasis of many carcinomas by binding to mesothelin in recipient tissues [26,27]. Recently it has been shown that mesothelin binds MUC16 thru a high affinity, for the latter [26,30,31]. Because a number of adenocarcinomas express MUC16, and adenocarcinomas represent one of the most common carcinomas metastasizing to the leptomeninges, it is conceivable that their interaction with meningioma mesothelin also facilitates anchoring there. Future studies may clarify the relative risk of specific carcinoma subtypes in spreading to the leptomeninges and mengiomas at specific regions of the brain. The relative importance of various adhesion molecules and identification of molecules blocking those interactions may in some cases, proactively limit metastases.

## Figures and Tables

**Figure 1 cancers-14-05877-f001:**
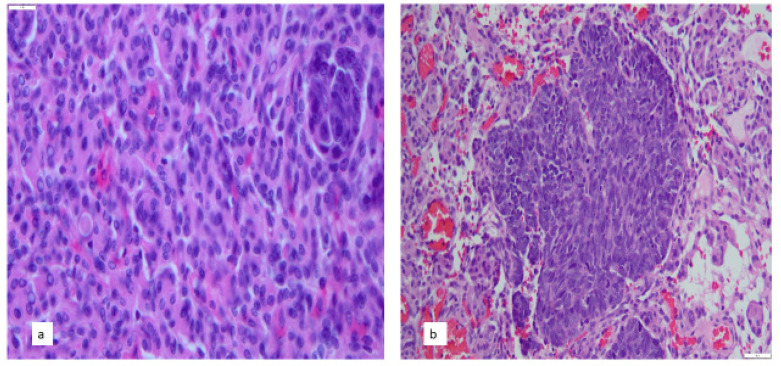
(**a**) Small cell carcinoma metastasis to meningioma (upper right corner) (**b**) Larger focus of metastatic small cell carcinoma in the meningioma. ((**a**) Hematoxylin and eosin original magnification 400× and (**b**) Hematoxylin and eosin original magnification 200×).

**Figure 2 cancers-14-05877-f002:**
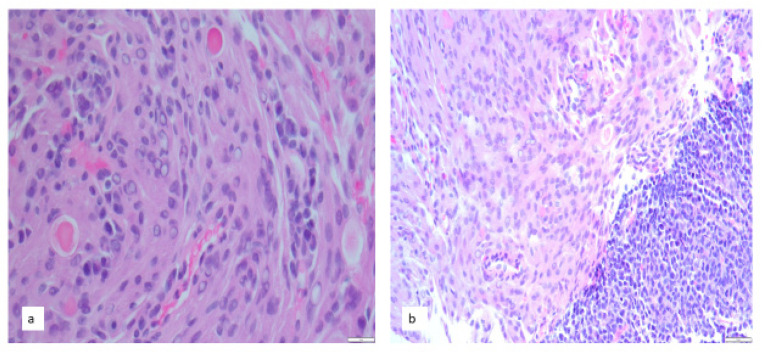
Metastatic plasma cell neoplasm to meningioma. (**a**) Foci of plasma cells (right) in sections of meningioma may suggest a lymphoplasmacytic variant unless, (**b**) the sample is completely submitted. All figures: Hematoxylin counterstain (Hematoxylin and eosin, Original magnification 400×).

**Figure 3 cancers-14-05877-f003:**
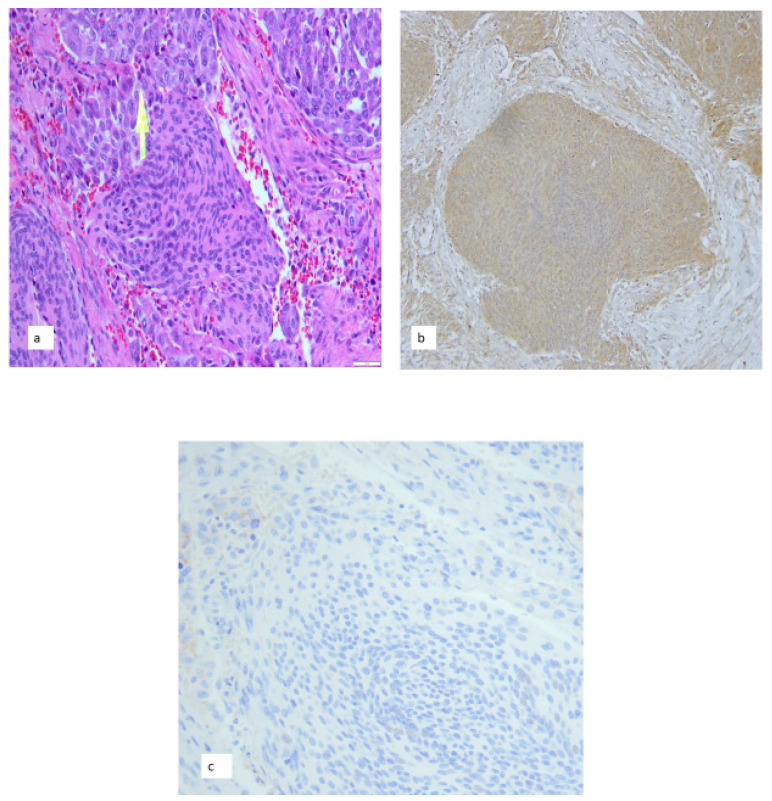
(**a**) Meningioma, WHO grade 2, with metastatic adenocarcinoma (arrow). (**b**) Same meningioma with mesothelin immunoreactivity and (**c**) MUC16 in adenocarcinoma (top) metastatic to the meningioma. (Hematoxylin and eosin original magnification 400×) (**a**) Hematoxylin counterstain. and diaminobenzidine chromagen. Original magnification 400×) (**b**,**c**).

**Table 1 cancers-14-05877-t001:** Metastatic carcinoma to meningioma.

	Carcinoma Site	Percentage	Age/Gender	Ref.
1	Breast	46%	59Y/100% F	[10]
2	Lung Adenocarcinomas	26%	61Y/55% F	[9]
3	Kidney	11%	57Y/44% F	[10]
4	Skin	1%	63Y/0 F	[10]

## Data Availability

The data presented in this study are available on request from the corresponding author.

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
