# Peer review of "Metastases to Meningiomas: A Comprehensive Literature Review Including Mediating Proteins"

_cancers, 2022, doi:10.3390/cancers14235877_

Round 1

Reviewer 1 Report

Dr. Johnson writes about the metastases to meningiomas, together with a literature review and several case examples to support the salient points, while discussing the molecular basis of this process. This is a good review and and analyzes the subject from a more basic scientific perspective. Although metastasis to meningioma is well-recognized, this review provides a rare recent reiteration from this perspective, as well. 

My main criticism, with the hopes of improving this manuscript, is regarding the format. The manuscript requires rigorous editing, formatting, and typesetting. It appears as though it was written in a whim and can benefit greatly from re-organization. A few examples include reference numbers in the abstract, reference numbers being limited/omitted in Table 1 relative to those cited in the related text, the entire paragraph right before Figure 1 being too confusing with incomplete sentences, the legend for Figure 3 being somewhat jumbled up, etc.

The Conclusion appears to be too long and includes sentences that have already been mentioned in the main text. It should be shortened to just give the reader the bottom-line. 

The figures and references are adequate and appropriate.

One question I have is on whether metastases have any relation to meningioma grade, since it is mentioned that higher grade meningiomas are more vascular. 

Author Response

//

Responses

1) The reference number has been removed from the abstract.

2) References have been added to Table 1

3) The paragraph preceeding Fig. 1 has been rewritten. The designation “(Fig. 1) “ has been moved to preceed mention of the lung finding.

3) The legend for Fig 3 has been changed for clarity.

Comment: The Conclusion appears to be too long and includes sentences that have already been mentioned in the main text. It should be shortened to just give the reader the bottom-line. 

Response: Some redundancy in the conclusion have been deleted. Nonetheless,I respectfully suggest that some essential facts have been left in since I think many readers peruse the conclusion first.

Comment: The figures and references are adequate and appropriate.

Comment: One question I have is on whether metastases have any relation to meningioma grade, since it is mentioned that higher grade meningiomas are more vascular. 

Response: This is already mentioned in the manuscript top of  p.9

Reviewer 2 Report

This is an excellent review of metastases to meningiomas including discussion of molecular mechanisms involved. The manuscript is well written, scientifically sound and provides highly relevant information to the readers.

Would suggest minor modification of last paragraph of “Histopathology” section which has some minor  typographical errors and also possibly placing reference to Fig.1 after sentence “Within it, a small focus of small cell carcinoma with an immunophenotype favoring small cell carcinoma” as current version may suggest that Fig. 1 shows tissues from the lung biopsy.

Author Response

Comment: Would suggest minor modification of last paragraph of “Histopathology” section which has some minor  typographical errors and also possibly placing reference to Fig.1 after sentence “Within it, a small focus of small cell carcinoma with an immunophenotype favoring small cell carcinoma” as current version may suggest that Fig. 1 shows tissues from the lung biopsy.

Response

The typographical errors have been corrected.

Figure 1 would not fit below the statement so is at beginning of next page.

Added in the legend it is in “the  meningioma” to make clear the focus is in the meningioma.